# The SARS-COV-2 Seroprevalence among Oncology Patients

**DOI:** 10.3390/jcm12020529

**Published:** 2023-01-09

**Authors:** Mankgopo Kgatle, Rajesh Das, Ismaheel Lawal, Tebatso Boshomane, Kgomotso Mokoala, Cattleya Gaspar, Lydia Mbokazi, Nonhlanhla Nkambule, Veronique Gow, Honest Ndlovu, Yonwaba Mzizi, Joseph Chalwe, Jeaneth Diphofa, Dinah Mokobodi, Nobuhle Gxekwa, Lusanda Zongo, Tinashe Maphosa, Mariza Vorster, Sheynaz Bassa, Amouda Venkatesan, Richard Khanyile, Yunus Munga, Thomas Ebenhan, Jan Rijn Zeevaart, Mike Sathekge

**Affiliations:** 1Nuclear Medicine Research Infrastructure (NuMeRI), Steve Biko Academic Hospital, Pretoria 0001, South Africa; 2Department of Nuclear Medicine, University of Pretoria & Steve Biko Academic Hospital, Pretoria 0001, South Africa; 3Department of Bioinformatics, Pondicherry University, Kalapet, Puducherry 605014, India; 4Department of Nuclear Medicine, University of Pretoria, Pretoria 0001, South Africa; 5AXIM Nuclear & Oncology (Pty) Ltd., Laboratory & Scientific Division, 63 Old Pretoria Road, Midrand 1685, South Africa; 6Radiation Oncology Outpatient Department, Ground Floor, Steve Biko Academic Hospital, Pretoria 0001, South Africa; 7Medical Oncology, Outpatient Department, Ground Floor, Steve Biko Academic Hospital, Pretoria 0001, South Africa; 8Faculty of Health Science, University of Pretoria, Pretoria 0001, South Africa; 9Department of Dermatology, University of Pretoria & Steve Biko Academic Hospital, Pretoria 0001, South Africa; 10Dermatology Department, Ondangwa Private Hospital, Ondangwa Box 2775, Namibia; 11South African Nuclear Energy Corporation, Radiochemistry, Elias Motsoaledi Street, R104 Pelindaba, Brits 0240, South Africa

**Keywords:** IgG/IgM antibodies, COVID-19, cancer, SARS-CoV-2, South Africa

## Abstract

Patients with cancer are presumed to be vulnerable to an increased risk of severe acute respiratory syndrome coronavirus 2 (SARS-CoV-2) infection and severe clinical outcomes due to the immunocompromised state mediated by their underlying malignancies and therapy. The aim of this study was to estimate the SARS-CoV-2 seroprevalence, following second to fourth waves in solid tumour patients attending the Steve Biko Academic Hospital (SBAH) for diagnosis and treatment of cancer. We used the single-prick COVID-19 IgG/IgM Rapid Test Cassettes to detect SARS-CoV-2 IgG/IgM antibodies in 760 patients with solid tumours who were asymptomatic and who had never tested positive for coronavirus disease 2019 (COVID-19). Out of the 760 patients, 277 were male (36.4%), 483 were female (63.6%), and the mean age was 55 years (range 18–92). The estimated total seroprevalence was 33.2%. The seroprevalence status of the COVID-19 IgG/IgM antibodies rose significantly from the second wave (11.3%) to the third (67.38%) and then the fourth (69.81%) waves with roughly similar counts. A significant number of the seropositive patients were asymptomatic to COVID-19 (96%). There was a higher rate of seropositivity in cancer patients with hypertension (*p* < 0.05). Patients with breast, gynaecologic, and prostate cancers exhibited increased SARS-CoV-2 seropositivity. Although oncology patients may be susceptible to SARS-CoV-2 infection, our data indicate that these patients remained asymptomatic throughout various waves with an overall COVID-19 IgG/IgM antibody seropositivity of 33.16%, suggesting no risk of severe or fatal cases of COVID-19.

## 1. Introduction

The coronavirus disease of 2019 (COVID-19) is a highly infectious disease that is caused by severe acute respiratory syndrome coronavirus 2 (SARS-CoV-2) virus. It has caused the current epidemic, and the World Health Organisation (WHO) reports that it is responsible for over 6 million fatalities globally. The initial detection of this virus was in Wuhan, China, around December 2019, and it is now present in approximately 200 countries [1]. SARS-CoV-2 has been reported to have phylogenetic similarities to SARS-CoV-1 which was responsible for the SARS outbreak of 2002–2004 [2]. Infection by SARS-CoV-2 is characterized by rapid human-to-human transmission, and the host immune response plays a crucial role in disease pathogenesis and symptoms [3].

Following the declaration of COVID-19 as a pandemic by the WHO [4], numerous reports have indicated the need for a focus on the impact of this disease on cancer patients. These reports from clinical studies have shown that patients with cancer are more susceptible to COVID-19 than individuals without cancer, due to malignancy and treatments which lead to an immunosuppressive state [5,6]. South Africa was severely affected by the SARS-CoV-2 pandemic waves caused by the original strain of SARS-CoV-2 and three variants of concern (VOCs) (beta, delta, and omicron). Distinctively, South Africa had the initial surge for two of the five VOCs identified to date—namely, beta and omicron [7]. Furthermore, several cancer surveillance systems have been implemented for the collection of cancer data, but underreporting of prevalence and mortalities is still a challenge in South Africa [8]. Some individuals, including cancer patients, may remain asymptomatic following SARS-CoV-2 infection, and they may continue to further spread the infection unknowingly. It is therefore important to estimate the prevalence of SARS-CoV-2 exposure in cancer patients, and its effects on their clinical outcome. This will provide a better understanding of the impact of SARS-CoV-2 on cancer patients, and enable the implementation of effective strategies to lower the risk of becoming infected. Additionally, this will mitigate negative effects of the COVID-19 pandemic on the diagnosis, and offer better cancer care delivery. IgG and IgM antibodies to COVID-19 are usually produced by the body’s immune system to fight infection, and can be detected within 1-3 weeks after exposure. Serologic tests are effective tools for the rapid monitoring of previous infection, by detecting antibodies specific to the virus in individuals who may have been exposed to the virus and who remained asymptomatic or mildly infected. In this study, we employed single-prick COVID-19 IgG/IgM Rapid Test Cassettes (Zheijiang Orient Gene Biotech, China) to trace SARS-CoV-2 previous infection in cancer patients treated at the Steve Biko Academic Hospital (SBAH), South Africa, where no specific study of this kind had been performed previously, according to our knowledge. The sensitivity of the single-prick COVID-19 IgG/IgM Rapid Test is 87.9% for IgM and 97.2 for IgG, as compared to RT-PCR. The specificity for both IgM and IgG antibodies is 100%.

## 2. Materials and Methods

### 2.1. The Characterisation and Location of the Study

This is a single-centred prospective and exploratory study conducted in patients visiting Radiation and Medical Oncology departments at the SBAH. The SBAH is a 900-bed quaternary academic hospital that serves as the referral centre providing oncology care for patients from four out of nine provinces in South Africa. Radiation and Medical Oncology departments are responsible for the management of both in- and out-patients with various cancers. They are about a 500 m bridge away from the SBAH complex, a combination of the SBAH and Tshwane District Hospital (TDH) bridge, that was repurposed for the management of COVID-19 patients.

### 2.2. Patient Population

Cancer patients who are newly diagnosed or receiving treatment for solid tumours at the Radiation and Medical Oncology departments of SBAH were prospectively recruited for this study. From 24 March 2021 to 30 March 2022, following the second to fourth waves of COVID-19, 760 tests were performed in the Radiation and Medical Oncology departments using the single-prick COVID-19 IgG/IgM Rapid Test Cassettes (Zheijiang Orient Gene Biotech, Huzhou, China). The second wave occurred between November 2020 and January 2021, the third wave occurred between May 2021 and September 2021 and the fourth wave occurred between November 2021 and present day. The participants recruited were 760 patients with various solid tumours who were admitted or who visited the hospital for their cancer management. To validate the accuracy of the test, 150 non-cancer patients hospitalised due to COVID-19, with similar underlying co-morbidities as cancer patients, were also tested, and these tests were all seropositive as expected. These patients were recruited from the same hospital for another unpublished study. The study was also approved by the University of Pretoria (Ethics Ref No: 28/2021).

### 2.3. Protocol for IgG/IgM Antibody Detection

The detection of SARS-CoV-2 IgG/IgM antibodies was carried out using the single-prick COVID-19 IgG/IgM Rapid Test Cassettes according to the manufacturer’s instructions (Zheijiang Orient Gene Biotech, China). This test is specific for detecting SARS-CoV-2 anti-nucleocapsid antibody. It does not detect anti-spike antibody, hence patients who have been vaccinated for COVID-19, who have not been previously infected by the wild virus, do not test positive on the rapid test kit. A test cassette was removed from the sealed foil pouch and used immediately. Briefly, a test cassette was placed on a clean and level surface. The participant’s finger was pricked with a needle and a drop of blood was drawn with a plastic dropper and transferred to the specimen well of a test cassette. Two drops of sample buffer, provided with the kit, were added immediately into the buffer well on a test cassette. The result was ready within 10 min. In the vast majority of tests, positive results became visible within 2 min.

### 2.4. Statistical Analysis

Statistical calculations were performed using the IBM^®^ SPSS^®^ Statistical Package for the Social Sciences (SPSS) version 28 (IBM, Armonk, NY, USA). A comparison between categorical variables was computed using Fisher’s exact test or Pearson Chi-squared test. A *p*-value of <0.05 was considered statistically significant. To calculate the seroprevalence, the following formula was used:(1)$$\mathrm{Seroprevalence}=\frac{No\;of\;patients\;with\;characteristic\;\left(positive\;or\;negative\right)}{Total\;number\;of\;patients\;included\;in\;the\;study\;\left(n=760\right)}$$

The percentage of the total cumulative number of COVID-19 cases and number of people vaccinated with either Johnson & Johnson or Pfizer vaccines, that were reported for both Gauteng Province and nationally (South Africa), was obtained from the daily reports of South African coronavirus latest statistics website on the 5 August 2022 [9]. These data were used to compare seropositivity and vaccination percentage among the population of South Africa, Gauteng Province, and our cancer patients’ data.

Age (<40, 40–69, ≥70 years), gender (female and male), smoking status (yes, no, ex-smoker), co-morbidities such as HIV, hypertension, diabetes, cancer type (breast cancer, colon and rectal, gynaecologic cancer, prostate cancer, and others), and COVID-19 symptoms (yes/no) were evaluated in order to include them in a binary logistic regression analysis model. Because seropositivity is a dependent variable that is dichotomous/binary in nature, basic linear regression could not be applied. Instead, binary logistic regression was used for dichotomous or binary dependent variables. The factors were found to be significant at a relaxed level (*p* < 0.6). The model’s goodness-of-fit was assessed using the Hosmer–Lemeshow test. Odds ratios (OR) and associated 95% confidence intervals (CI) were generated to determine the degree of association.

## 3. Results

### 3.1. General Characteristics

In this study, a total of 760 cancer patients were included (Figure 1). A total of 150 non-cancer patients hospitalised with COVID-19 were also included for the validation of the test, and all of them were COVID-19 IgG/IgM seropositive as expected (data not shown). The ratio of male to female, favoured females, and the average age of the study group was 55 years (range: 18 to 92). Of the 760 patients, 277 (36.4%) of them were male, and 483 (63.6%) were female. There was greater participation of non-smokers (84.1%) and frequencies of patients with co-morbidities such as HIV (19.6%), hypertension (20.4%), and diabetes (20.8%).

Almost three-quarters of the treatments were radical treatment (72%); the rest was palliative (17.5%) or other (10.5%). Overall, 322 (42.4%) out of 760 patients were previously under cancer treatment (13.3% chemotherapy, 11.3% radiotherapy, 28.7% surgery, 0.7% Zoladex, and 57.8% either completed or other therapy). A total of 386 (50.8%) patients were on active cancer treatment (34.9% chemotherapy, 28.8% radiotherapy, 0.3% surgery, 1.2% Zoladex, and 39.3% either completed or had other therapy).

### 3.2. Seroprevalence of COVID-19

A more significant number of the seropositive patients were asymptomatic to COVID-19 (96%). Out of 760 patients, 153 (20.1%) tested positive for SARS-CoV-2 IgG antibodies, 4 (0.53%) for SARS-CoV-2 IgM antibodies, and 94 (12.5%) for both IgG and IgM antibodies. The estimated total seroprevalence (i.e., IgG or IgM positive) was 33.2%. A summary regarding the count of antibody test status is given in Table 1. The cancer patient’s age group, HIV status, and diabetes status were not significantly related to SARS-CoV-2 seropositivity (*p* > 0.05), as opposed to gender, smoking, different cancers, hypertension, and symptoms that exhibited a significant relationship with SARS-CoV-2 seropositivity (*p* < 0.05) (Table 2). From our analysis, we have observed that the percentage of female cancer patients is higher than of males, irrespective of the seroprevalence status (Table 2). Among the seropositive patients, it is observed that 57.94% of the cohort are females. Furthermore, 66.34% of the screened seronegative cancer patients were females. In the case of treatment, previous cancer treatment, radiotherapy, and Zoladex monotherapy did not have significance in the pandemic. Again, in the current cancer treatments, Zoladex monotherapy with surgery did not have a statistical significance on seropositivity (Table 3).

With respect to the second wave, the seropositivity increased by almost six-fold to 67.38% and 69.81% for the third and fourth waves, respectively (Figure 2A). The seropositivity was 11.13% for the second wave. This information means that the negativity rate was higher in the second wave, while the positivity rate increased in the third and fourth waves. The percentage population infected with COVID-19 in South African national data and Gauteng Province data has also been calculated and is shown in Figure 2B. In national data, the seropositivity was found to be 2.99%, 4.98%, and 6.11% for second, third, and fourth waves, respectively. In Gauteng Province, the total cumulative cases were 26.8%, 31.5%, and 32.5% for second, third, and fourth waves, respectively.

### 3.3. Vaccination in the Population

The information for Johnson & Johnson and Pfizer vaccination was collected for national (South Africa) and Gauteng Province from the daily reports of South African coronavirus latest statistics website on the 5 August 2022 [9] and this was compared with our data from cancer patients. The comparison of vaccination percentage can be seen in Figure 3.

In the second wave, the percentage of vaccination is comparatively less with respect to third and fourth waves. In the second wave, the vaccination percentage was 0.53%, 0.71%, and 0.42% for national, Gauteng Province, and our cancer patients’ data, respectively. In the third wave, 25.49%, 35.22%, and 55.08% of the people were vaccinated for national, Gauteng Province, and our cancer patients’ data, respectively. Finally, in the fourth wave, vaccination percentage was 32.45%, 46.43%, and 47.16% for national, Gauteng Province, and our cancer patients’ data, respectively. It appears that the vaccination percentage increased with waves, especially for the national and Gauteng Province, as compared to our cancer patients’ data.

### 3.4. The Regression Model

The omnibus test showed a *p*-value of <0.001, indicating that the regression model was a good fit compared to the null model. The goodness-of-fit test showed low significance (*p* > 0.4). There was no difference between the observed and predicted models. The Nagelkerke R Square value for the model was 0.311, suggesting that 31.1% of the dependent variable could be accounted for by the dependent variables of the model. The classification table indicates that the independent variables correctly predicted the seropositivity of the patients with a rate of 75.8%, which may be considered to measure the prediction accuracy of the model.

The odds ratio of every variable indicates the probability of falling into the target group and the non-target group. The odds ratio and 95% CI of the factors associated with seropositivity are presented in Table 4. Patients aged less than 40 years were more prone to COVID-19 than the age group of between 40–69 (95% CI; 0.419–1.19) and ≥70 (95% CI; 0.37–1.46). Female patients (95% CI; 0.40–1.48) were less likely to be seropositive than males. The number of non-smokers was higher in the population. Smokers (95% CI; 0.57–1.70) and non-smokers had an almost equal probability of being SARS-CoV-2 seropositive. According to the smoking history of the patients, ex-smokers (95% CI; 0.70–6.60) had 2.15 times higher chance of being infected with coronavirus, when compared to non-smokers. COVID-19 seropositivity prevalence was 1.25 times among HIV patients (95% CI; 0.78–2.01) compared to non-HIV patients. Patients with hypertension (95% CI; 1.18–2.80) had 1.8 times higher seroprevalence than patients without hypertension. Again, patients with diabetes mellitus (95% CI; 0.86–2.20) had approximately 1.4 times higher seroprevalence than those with no diabetes mellitus.

Cancer patients who were symptomatic (95% CI; 0.27–1.41) of SARS-CoV-2 had less seroprevalence than asymptomatic patients. The cancer types such as breast, colon and rectal, gynaecologic, and prostate cancers had higher seroprevalence when compared to other cancer types. Breast (95% CI; 0.99–3.68), gynaecologic (95% CI; 1.19–4.17), and prostatic cancers (95% CI; 1.29–4.84) had almost two times, or higher, seroprevalence than the other cancer types. Colon and rectal cancer (95% CI; 0.57–3.94) had 1.5 times higher seroprevalence than other cancer types. Radical (95% CI; 0.03–0.21) and palliative (95% CI; 0.16–0.57) treatments were negatively associated with COVID seroprevalence. There were two stages of treatments, namely previous and current cancer treatment. None of the previous cancer treatments showed higher seroprevalence. On the contrary, radiotherapy (95% CI; 0.46–3.35) and surgery (95% CI; 0.12–51.69) in current cancer treatments showed 1.2 and 2.5 times higher seroprevalence, respectively.

## 4. Discussion

Our study focused on tracking the seroprevalence of anti-SARS-CoV-2 IgG and IgM antibodies in patients with solid tumours following three successive COVID-19 waves with different VOCs including beta, delta, and omicron that occurred from November 2020 to January 2021, May 2021 to September 2021 and November 2021 to present, respectively, in South Africa [10]. Our data show that SARS-CoV-2 exposure in cancer patients rose significantly with the waves, and yet these patients remained asymptomatic. Prostatic, gynaecologic, and breast cancers exhibited higher seroprevalence than their reference category. These cancers are common in the South African population and clinical settings [11,12]. Approximately 50% of the total cancer patients tested for SARS-CoV-2 seroprevalence had breast and gynaecologic cancers, explaining the increased seroprevalence and favouritisms towards females.

Gender, smoking, different cancers, hypertension, and symptoms also exhibited significant relationships with SARS-CoV-2 seropositivity, and this may be attributable to various mechanisms previously postulated. Transmembrane protease serine 2 (TMPRSS2), angiotensin-converting enzyme 2 (ACE-2), and other host cell proteases including cathepsin L, which is frequently expressed in cancer patients, are known to enhance COVID-19 infection [13,14]. The immunosuppressive conditions of cancer patients render them more susceptible to serious COVID-19 consequences than the general population, which may have an impact on the disease’s prognosis. Several studies reported a consistent relationship between past smoking and elevated ACE-2 expression in a series of seminal in vitro experiments [6,15,16]. The association was dose-dependent, and also detectable in both bulk and single-cell analyses, which remained significant after multivariate linear regression controlling for factors such as age, sex, race, and body mass index [16]. Since smoking increases the expression of ACE-2, it is conceivable that smokers are exposed to a larger SARS-CoV-2 load. This could offer a mechanistic explanation for the increased risk of infection, as well as fatal illness and mortality linked to smoking in COVID-19 patients. According to Liang et al. [17], cancer patients are more likely to experience adverse outcomes such as the need for an intense level of care, mechanical breathing, and mortality (39% vs. 8%, *p* = 0.0003) than non-cancer patients. It is also noted that 70% of stage 4 cancer patients experienced serious events [18]. If any harsh immunosuppressive chemotherapy treatment is recommended [18], we suggest that the dose be reduced or postponed for the patients who are generally in poor condition.

The beta variant drove the second wave of SARS-CoV-2 infections occurring from November 2020, and it exited in January 2021 [10]. Our analysis demonstrated that this variant contributed the highest SARS-CoV-2 infection rates and was 57% more likely to cause severe to fatal cases than the earlier variant that contributed only 25%. During the early days of the COVID-19 pandemic, SBAH delayed treatments, and reduced the number of visits and appointments for oncology outpatients as part of COVID-19, prioritizing management and protective measures to contain the infection and protect vulnerable patients [10]. Additionally, strict adherence to social distancing and other measures were also put in place. This may explain an increased percentage of SARS-CoV-2 negative seroprevalence by oncology patients in the second wave, as observed in our data. Perceived increased SARS-CoV-2 infection susceptibility and associated aggressiveness of COVID-19 in oncology patients may have also encouraged these patients to avoid crowds and public spaces that may have put them at high risk to exposure. Indeed, many patients reported that they avoided unnecessary outings to protect themselves. Additionally, it was rather surprising to those who had tested positive for anti-SARS-CoV-2 antibodies because they were unaware that they had been exposed. Although we did not collect the data for the first wave, we believe that the trajectory may have been similar because of the early onset of the pandemic and the above protective measures.

South Africa started its national vaccination program on 21 February 2021 prioritizing healthcare/frontline workers and those over the age of 60 [19]. This was followed by the easing of some of the restrictions introduced above, explaining the shifted prevalence trajectory with a significant rise in seropositivity (67.4%) following the third wave. The third wave was dominated by the delta variant that occurred between May 2021 and September 2021, and it was associated with a surge of SARS-CoV-2 infections accompanied by higher numbers of fatalities [10]. This continued with the fourth wave that was dominated by the omicron variant, which was spreading twice as fast as the delta. The increased seropositivity prevalence trajectory may have been attributable to the biopsychosocial effects of vaccination. The subsequent ease of some protective measures initially put in place by the hospital, and established knowledge of COVID-19, may have encouraged less concern about exposure.

Most patients never tested positive for COVID-19, and many were unvaccinated at the time when they were recruited to participate in this study. About 35.22% of people in the Gauteng province had been vaccinated with at least one dose by the time of the third wave, and this increased to 46.4% during the fourth wave. This suggested an increase in waves as observed with the whole South African data, partly attributable to ongoing vaccination promotion campaigns. Although the same trajectory was observed from the second to third waves (0.4 to 55.1%), the percentage dropped down to 47.2% with the cancer patients’ data. This may have just been by chance or partially due to COVID-19 vaccine hesitancy, which has been a big challenge in South Africa [20]. Common reasons for this hesitancy included the perception that these vaccines were a government or global plot for making money, or the concern that these vaccines were unsafe or harmful as their production and roll-out were rushed without considering the profound testing of safety and side effects reported by early-on studies [21,22].

Interestingly, our data also show that most patients, regardless of vaccination status, were unaware and only found out through our screening that they were silently developing SARS-CoV-2, i.e., reporting that they had never experienced any symptoms or sickness suggesting any exposure to SARS-CoV-2 infection. This is similar to the findings obtained by other studies that also found their cancer populations to be asymptomatic to COVID-19 [23,24,25]. This suggests that, although these patients may be susceptible to SARS-CoV-2 infection, they remained asymptomatic throughout the pandemic, and thus they were not at risk of severe illness necessitating invasive ventilation and intensive care unit (ICU) hospitalisation, as several reports initially assumed [26,27,28,29,30]. It has been reported that possible pre-existing and de novo humoral immunity, that stemmed from previous coronavirus exposure, rendered some protection to children and individuals who are supposedly at risk due to human immunodeficiency virus and tuberculosis infection [31,32]. Ng et al.’s [31] study has demonstrated that 5% and 62% of SARS-CoV-2 unexposed adults and children, respectively, had antibodies that also recognise SARS-CoV-2. These antibodies appear to be produced through previous coronaviruses that share similar genome features with the SARS-CoV-2 virus. Indeed, the S2 subunit of SARS-CoV-2 is similar across other coronaviruses, and can stop SARS-CoV-2 virus from entering the cells of previously exposed individuals, and thus protecting them against the new exposure with SARS-CoV-2. This may also be the same case with the vast majority of oncology patients. Due to the immunocompromised status of oncology patients, it is possible that some of these patients have also been previously exposed to other coronaviruses, and this rendered them protection, explaining less exposure to SARS-CoV-2 or maintained asymptomatic or mild status when infected.

Our study has some limitations. For instance, due to the boundaries of the ethics approval and the single centre prospective and exploratory study method employed, our findings should not be generalized to the entire population, but only applied to the current population due to inadequate thoroughness. Furthermore, diagnosis was only based on the result of a single test, not in conjunction with SARS-CoV-2 PCR tests or complete clinical assays.

## 5. Conclusions

The asymptomatic presentation of SARS-CoV-2 viral infection in cancer patients is possible but uncommon. This occurrence presents a challenge to the health sector for both diagnosis and treatment of these patients. Our study supplements current information on the presentation of SARS-CoV-2 viral infection in cancer patients, which may be used by policy makers and healthcare providers to improve the management of these patients.

## Figures and Tables

**Figure 1 jcm-12-00529-f001:**
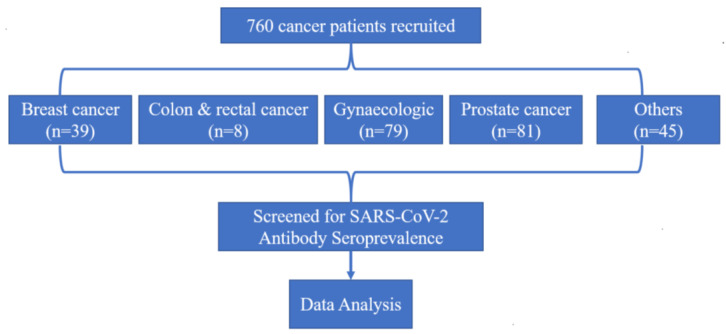
Study profile.

**Figure 2 jcm-12-00529-f002:**
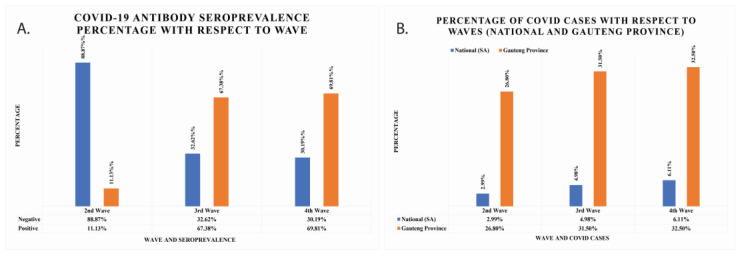
(**A**) The COVID-19 IgG/IgM seroprevalence percentage of cancer patients per wave represented with grey colours. The positive seroprevalence of COVID-19 IgG/IgM antibodies rises with waves. (**B**) The total number of COVID-19 cumulative cases per waves reported nationally and in Gauteng Province.

**Figure 3 jcm-12-00529-f003:**
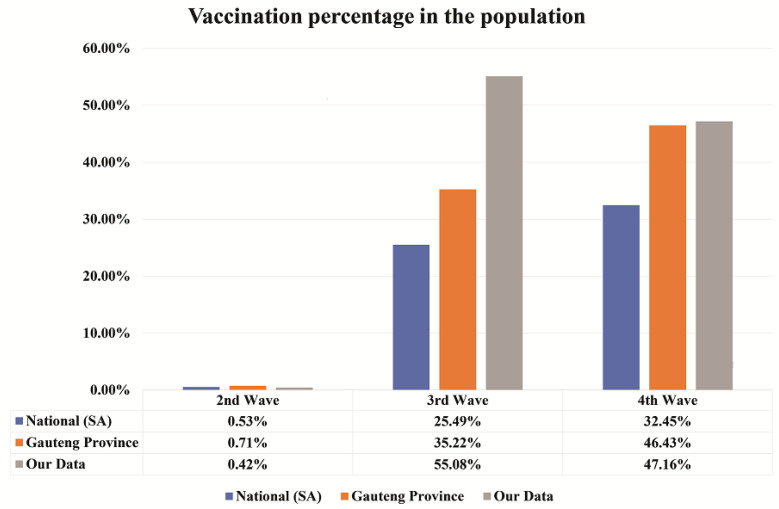
The vaccination percentage per wave. A comparison of vaccination percentage in the South African population, Gauteng Province, and our cancer patient’s data represented in blue, orange, and grey colours, respectively. Vaccination percentage is seen to be increased with waves. Our test is specific for detecting SARS-CoV-2 anti-nucleocapsid than anti-spike antibodies associated with vaccines.

**Table 1 jcm-12-00529-t001:** Count and percentage of SARS-CoV-2 seroprevalence in cancer patients.

	Count of Antibody Test Status	Percentage %	Overall Seropositivity %
IgG and IgM positive	95	12.50	Positive= 33.16
IgG positive	153	20.13	
IgM positive	4	0.53	
Negative	508	66.84	Negative= 66.84
Total	760		

Abbreviations: IgG—immunoglobulin G; IgM—immunoglobulin M.

**Table 2 jcm-12-00529-t002:** General characteristics of patients, tumour type, underlying co-morbidities, and SARS-CoV-2 seroprevalence.

	SARS-CoV-2 Seropositive		SARS-CoV-2 Seronegative		*p*-Value
	n (252)	%	n (508)	%	
**Age (years)**					0.27
<40	37	14.70	82	16.14	
40–69	163	64.70	345	67.91	
≥70	52	20.60	81	15.94	
**Gender**					0.023
Male	106	42.06	171	33.66	
Female	146	57.94	337	66.34	
**Smoker**					0.009
Yes	32	12.70	71	13.98	
No	208	82.54	431	84.84	
Ex-smoker	12	4.76	6	1.18	
**Co-morbidities**					
**HIV**					0.14
Yes	57	22.62	92	18.11	
No	195	77.38	416	81.89	
**Hypertension**					<0.001
Yes	89	35.32	66	12.99	
No	163	64.68	442	87.01	
**Diabetes**					0.941
Yes	52	20.63	106	20.87	
No	200	79.37	402	79.13	
**Cancer type**					<0.001
Breast Cancer	39	15.48	158	31.10	
Gynaecologic Cancer	79	31.35	83	16.34	
Colon and Rectal	8	3.17	32	6.30	
Prostate	81	32.14	82	16.14	
Others	45	17.86	153	30.12	
**Symptoms**					0.009
Yes	9	3.57	44	8.66	
No	243	96.43	464	91.34	

**Table 3 jcm-12-00529-t003:** Characteristics of cancer treatment for the screened patients and SARS-CoV-2 seroprevalence.

	SARS-CoV-2 Seropositive		SARS-CoV-2 Seronegative		*p*-Value
	n (252)	%	n (508)	%	
**Treatment Intent**					<0.001
Radical	191	75.79	356	70.08	
Palliative	10	3.97	123	24.21	
Others ^a^	51	20.24	29	5.71	
**Previous Cancer Treatment**					
**Chemotherapy**					<0.001
Yes	16	6.35	85	16.73	
No	236	93.65	423	83.27	
**Radiotherapy**					0.397
Yes	32	12.70	54	10.63	
No	220	87.30	454	89.37	
**Zoladex**					0.531
Yes	1	0.40	4	0.79	
No	251	99.60	504	99.21	
**Surgery**					<0.001
Yes	40	15.87	178	35.04	
No	212	84.13	330	64.96	
**Others** ^b^					<0.001
Yes	174	69.05	265	52.17	
No	78	30.95	243	47.83	
**Current Cancer Treatment**					
**Chemotherapy**					<0.001
Yes	34	13.49	231	45.47	
No	218	86.51	277	54.53	
**Radiotherapy**					<0.001
Yes	100	39.68	119	23.43	
No	152	60.32	389	76.57	
**Zoladex**					0.469
Yes	4	1.59	5	0.98	
No	248	98.41	503	99.02	
**Surgery**					0.612
Yes	1	0.40	1	0.20	
No	251	99.60	507	99.80	
**Others** ^b^					<0.001
Yes	122	48.41	177	34.84	
No	130	51.59	331	65.16	

^a^ Treatment intent other than radical and palliative is termed as others.; ^b^ Cancer treatment other than chemotherapy, radiotherapy, Zoladex, or surgery is considered under “Others” category, as the number of patients having these cancer types is less.

**Table 4 jcm-12-00529-t004:** Binary logistic regression analysis of factors associated with seropositivity.

	Odds Ratio (OR)	95% CI	
		Lower	Upper
**Age Group**			
<40	Ref		
40–69	0.706	0.419	1.189
≥70	0.728	0.365	1.452
**Gender**			
Male	Ref		
Female	0.773	0.404	1.479
**Smoker**			
No	Ref		
Yes	0.987	0.574	1.699
Ex-smoker	2.154	0.704	6.591
**Co-morbidities**			
HIV			
No	Ref		
Yes	1.251	0.778	2.011
**Hypertension**			
No	Ref		
Yes	1.820	1.181	2.804
**Diabetes**			
No	Ref		
Yes	1.373	0.856	2.203
**Cancer type**			
Others	Ref		
Breast Cancer	1.908	0.988	3.683
Colon and Rectal Cancer	1.532	0.596	3.939
Gynaecologic Cancer	2.224	1.187	4.166
Prostate Cancer	2.500	1.290	4.844
**Symptoms**			
No	Ref		
Yes	0.661	0.265	1.407
**Treatment Intent**			
Others/None	Ref		
Radical	0.081	0.031	0.210
Palliative	0.303	0.161	0.571
**Previous Cancer Treatment**			
**Chemotherapy**			
No	Ref		
Yes	0.471	0.220	1.010
**Radiotherapy**			
No	Ref		
Yes	0.752	0.316	1.786
**Zoladex**			
No	Ref		
Yes	0.144	0.013	1.556
**Surgery**			
No	Ref		
Yes	0.411	0.179	0.943
**Others/Completed/None**			
No	Ref		
Yes	0.668	0.284	1.570
**Current Cancer Treatment**			
**Chemotherapy**			
No	Ref		
Yes	0.435	0.180	1.051
**Radiotherapy**			
No	Ref		
Yes	1.237	0.457	3.351
**Zoladex**			
No	Ref		
Yes	0.853	0.167	4.353
**Surgery**			
No	Ref		
Yes	2.522	0.123	51.689
**Others/Completed/None**			
No	Ref		
Yes	0.956	0.325	2.814

## Data Availability

Data supporting reported results can be requested by email from both corresponding authors.

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
