# Peer review of "The SARS-COV-2 Seroprevalence among Oncology Patients"

_jcm, 2023, doi:10.3390/jcm12020529_

Round 1

Reviewer 1 Report

Although this is a well-written article and addresses some interesting points, e.g about asymptomatic seropositivity, overall, its does not add to the current knowledge and research priorities about the pandemic. The results of the seroprevalence among oncology patients in a certain hospital during the previous waves do not make an impact on the management or national health policy priorities at this stage of the pandemic. The results have some value but they cannot be easily generalized.

Author Response

We appreciate this reviewer’s constructive criticism. We wish that they have been more specific about what exactly the problems were with the manuscript so that we can be able to address the comments. Unfortunately, this limited our ability to give the response point-by-point as requested. We hope that excellent comments provided by the second reviewer and the response we have given may provide more clarification. Additionally, we would like to emphasise that this manuscript does provide some important insights on the impact of COVID-19 pandemic especially when cancer patients are concerned. We are, therefore, now aware that cancer patients are not largely at risk of severe cases and fatal illness as it was proposed by many studies with no substantial data.

Reviewer 2 Report

Thank you for allowing me to review this manuscript. The manuscript is interesting and well written. Kindly find my comments below

• Abstract

o The percentage of seroprevalence among patients should be mentioned as the first result in the abstract, since this is the main goal of the study.

• Introduction

o The authors mentioned that they used single- 73 prick COVID-19 IgG/IgM Rapid Test Cassettes. What is the sensitivity and specificity of this test?

• Materials and Methods

o Would you please add the dates of each of the waves of COVID-19 you are discussing?

o You mentioned that “150 non-cancer patients hospitalized due to COVID-19 with similar 96 underlying comorbidities as cancer patients were also tested, and these were all seropositive as expected.” Where were these recruited from?

• Results

o The percentage of seroprevalence among patients should be mentioned as the first result in the section, since this is the main goal of the study.

o Please add more details in the text about the associations found in table 1

o What is Zoladex monotherapy used for? Why it is a specific category?

o The figures are not clear. Please increase their resolution.

o What are the types of vaccines that were used?

Discussion

• The first paragraph of the discussion includes repetition of numbers mentioned in the results. Please add a summary of your important findings without these numbers.

• You mentioned that “These cancers are common in the South African population and clinical setting, explaining this observation and favouritisms towards females.” Please explain more. How can the pattern or spread explain this observation?

• This section should include more discussions about the findings found in the results. For example, why was the seroprevalence different in different types of cancer? The effect of smoking? The effect of comorbidities and treatment? Even if there is no clear explanations for some findings, the authors may suggest some explanations based on their knowledge of the community.

Author Response

Dear Reviewer 

We would like to thank you for your valuable comments and inputs for the revision of this manuscript. We have addressed all the comments per your recommendation and incorporated the suggested information in the manuscript. We believe that this has hugely improved our manuscript.  

Please see the response on the attachment.

Round 2

Reviewer 1 Report

The manuscript has been significantly improved and therefore, i recommend it for publication.